# TsRNA-49–73-Glu-CTC: A promising serum biomarker in non-small cell lung cancer

**Chenyu Li[1,2], Shenjie Zhong[2], Juan Chen[2], Xiaofeng Mu[2]\***

**1** Qingdao Medical College, Qingdao University, Qingdao, Shandong, China, **2** Medical Laboratory, Qingdao Central Hospital, University of Health and Rehabilitation Sciences, Qingdao, Shandong, China

\* muxiaofeng2005@126.com

## Abstract

### Objective

Lung cancer has the highest incidence and mortality rates globally, with the majority of cases classified as non-small cell lung cancer (NSCLC). Due to the absence of specific tumor biomarkers, most lung cancer cases are diagnosed at an advanced stage. Therefore, the identification of novel molecular biomarkers with high sensitivity and specificity for early diagnosis is deemed crucial for enhancing the treatment of NSCLC. Transfer RNA-derived small RNA (tsRNA) is closely associated with malignant tumors and holds promise as a potential biomarker for tumor diagnosis. This study aimed to investigate whether serum tsRNA could serve as a biomarker for NSCLC.

### Methods

Differentially expressed tsRNAs were identified through high-throughput sequencing of serum samples obtained from patients with NSCLC and healthy individuals. Additional serum samples were collected for validation using Reverse Transcription Quantitative Polymerase Chain Reaction (RT-qPCR). The diagnostic performance of these tsRNAs was assessed through Receiver Operating Characteristic (ROC) Curve Analysis. Furthermore, preliminary functional exploration was undertaken through cell experiments.

### Results

tsRNA-49-73-Glu-CTC is highly expressed in the serum of patients with NSCLC and demonstrates superior diagnostic value compared to commonly used tumor markers in clinical practice, such as Carcinoembryonic Antigen (CEA), Neuron-Specific Enolase (NSE), and Cytokeratin 19 Fragment (CYFRA). A combined diagnostic approach enhances the accuracy of NSCLC detection. Additionally, tsRNA-49-73-Glu-CTC is highly expressed in A549 cells, and transfection with a tsRNA-49-73-Glu-CTC inhibitor significantly reduces both proliferation and migration capabilities.

**Data availability statement:** All data related to this study are included in the manuscript and

supporting information. The high-throughput sequencing data supporting the findings of this study are publicly available in the GEO database under accession number GSE278659, which can be accessed at https://www.ncbi.nlm.nih.gov/geo/query/acc.cgi?acc=GSE278659.

**Funding:** This study was supported by Qingdao Science and Technology Benefiting the People Demonstration Project (24-2-8-smjk-12-nsh) awarded to XFM and Qingdao Medical and Health Key Discipline Construction Project (2022-6) awarded to XFM.

**Competing interests:** The authors have declared that no competing interests exist.

## Conclusions

tsRNA-49-73-Glu-CTC has the potential to serve as a novel molecular diagnostic biomarker for NSCLC and plays a significant role in the biological processes associated with NSCLC proliferation and migration.

## 1 Introduction

According to global cancer statistics, lung cancer is the most common cancer worldwide [1,2]. Recent data indicate that lung cancer ranks first globally in terms of new cases, with an estimated 2.5 million diagnoses annually [2]. It is also the leading cause of cancer-related deaths, accounting for approximately 1.8 million fatalities, or 18.7% of all cancer deaths—far surpassing colorectal cancer, which ranks second at 9.3% [2,3]. The five-year survival rate for lung cancer remains below 20% [4–6]. NSCLC and small cell lung cancer (SCLC), with NSCLC comprising over 85% of cases [2]. Among NSCLC subtypes, adenocarcinoma is the most prevalent, accounting for approximately 40% of cases, followed by squamous cell carcinoma, which accounts for about 25% [6,7]. The lack of specific diagnostic biomarkers often leads to lung cancer being diagnosed at an advanced stage [8,9]. CEA, a commonly used clinical cancer biomarker, shows relatively high sensitivity for NSCLC, particularly adenocarcinoma [10–13]. However, as a broad-spectrum biomarker, its specificity is limited, as elevated levels can also be observed in other malignancies and some benign conditions [14,15]. Squamous Cell Carcinoma Antigen (SCCA) provides diagnostic value for advanced-stage lung squamous cell carcinoma but exhibits low sensitivity for early-stage lung cancer [3,16]. Pro-Gastrin-Releasing Peptide (Pro-GRP) is a specific biomarker for SCLC, with a sensitivity of 60%-70% for early-stage SCLC; however, it has no diagnostic utility for NSCLC [17,18]. Although tissue biopsy remains the gold standard for lung cancer diagnosis, invasive procedures can cause significant discomfort to patients and carry risks such as infection, pneumothorax, and other iatrogenic injuries. Imaging studies, particularly low-dose computed tomography (CT), have demonstrated diagnostic value in lung cancer [19]. However, their high cost and limited accessibility impede widespread adoption. Despite the reduced radiation dose, repeated screenings may result in cumulative radiation exposure, thereby increasing patients' overall radiation risk. Therefore, finding safer and non-invasive diagnostic methods is of great importance. This study aims to explore the potential of tsRNA as a diagnostic biomarker for lung cancer. With the advancement of high-throughput sequencing, an increasing number of non-coding RNAs have come into focus. Initially regarded as mere degradation products of tRNA [20]. tsRNAs have since been shown to play significant roles in various biological processes, including the inhibition of translation, regulation of mRNA function, and mediation of intercellular communication [21–23]. tsRNAs are primarily categorized into two types: tiRNA halves (tiRNA) generated by angiogenin (ANG) cleavage of the mature tRNA anticodon loop, and tRNA-derived fragments (tRFs) produced by ribonuclease (RNase) cleavage of precursor or mature tRNA ends [24]. tiRNAs typically range from 31–40 nucleotides in length, with both 5'-tiRNA and 3'-tiRNA types, while tRFs are shorter, usually 14–30 nucleotides in length, and can be cleaved from different regions of tRNA [25,26]. The surface of tsRNAs is modified in multiple ways, which confers stability in diverse body fluids, thereby positioning them as potential molecular diagnostic biomarkers.

Research has demonstrated that tsRNAs exhibit stable expression under normal physiological conditions; however, their expression levels become dysregulated in response to stress [27,28]. The expression profiles of tsRNAs vary across different diseases and cancers, and may also change depending on the stage of the disease or malignancy. For example, 5'-tiRNA-Cys-GCA

expression is downregulated in patients with aortic dissection [29]. tRF-20-S998LO9D is expressed at low levels in patients with endometrial cancer [30]. tRF-23-Q99P9P9NDD, highly expressed in the serum of gastric cancer patients, effectively distinguishes between gastric cancer and healthy individuals, as well as between benign gastric conditions and different stages of gastric cancer [31]. tiRNA-1:33-Pro-TGG-1 can assist in the diagnosis of sessile serrated lesions, contributing to the early prevention of colorectal cancer [32]. tsRNAs are stable in serum, and serum tsRNA can serve as valuable non-invasive biomarkers for pancreatic, gastric, and liver cancers [33–35]. Research has demonstrated through data analysis that certain tsRNAs in plasma could effectively differentiate lung adenocarcinoma patients from healthy individuals, indicating the potential of tsRNAs as diagnostic biomarkers for lung cancer. However, the expression and diagnostic value of tsRNAs in the serum of lung cancer patients remain inadequately understood and warrant further investigation.

The primary objective of this study was to investigate the relationship between serum tsRNA and NSCLC. Initially, high-throughput sequencing analysis of serum samples from three pairs of NSCLC patients and healthy individuals identified tsRNA-49-73-Glu-CTC as significantly differentially expressed. This finding was subsequently validated through RT-qPCR on the serum of 32 patients and 20 healthy individuals, confirming that tsRNA-49-73-Glu-CTC is highly expressed in the serum of NSCLC patients. ROC curve analysis revealed that tsRNA-49-73-Glu-CTC has a greater diagnostic value for NSCLC than CEA, NSE, and CYFRA. Moreover, a combined diagnostic approach significantly enhances diagnostic accuracy, suggesting that tsRNA-49-73-Glu-CTC may serve as a specific molecular diagnostic biomarker for NSCLC. Furthermore, the inhibition of tsRNA-49-73-Glu-CTC expression reduced the proliferation and migration abilities of A549 cells, indicating that tsRNA-49-73-Glu-CTC may play a role in the proliferation and migration processes associated with NSCLC.

## 2 Methods and materials

### 2.1 Samples and cells

All serum samples utilized in this study, which were gotten from 32 NSCLC patients and 20 healthy controls from 20/09/2024 to 30/10/2024, were sourced from the Qingdao Central Hospital, University of Health and Rehabilitation Science (Qingdao Central Hospital). The NSCLC patients included in the study were pathologically confirmed, had not undergone any treatment at the time of serum collection, and provided informed consent in accordance with ethical guidelines. Serum samples were collected and stored following standard procedures. The clinical information of the patients is provided in S1 Table.

The human bronchial epithelial cell line (BEAS-2B) cells served as a control to simulate normal lung epithelial cells, while A549 cells were chosen because lung adenocarcinoma was the predominant pathological type observed among patients, and A549 is a commonly used lung adenocarcinoma cell line. BEAS-2B cells were obtained from FuHeng Biology (Shanghai, China). A549 cells were obtained from Pricella Biotechnology (Wuhan, China). BEAS-2B cells were cultured in high-glucose DMEM medium (ThermoFisher Scientific, MA, USA) supplemented with 10% fetal bovine serum (FBS) (ThermoFisher Scientific, MA, USA) and 1% penicillin/streptomycin. A549 cells were cultured in Ham's F-12K medium (ThermoFisher Scientific, MA, USA) supplemented with 10% FBS and 1% penicillin/streptomycin. Cells were maintained in a 37°C incubator with 5% $CO_2$.

### 2.2 Sequencing with high throughput

Total RNA (extracted from the serum of 3 NSCLC patients and 3 healthy controls) are qualified by agarose gel electrophoresis and quantified using Nanodrop™ instrument. Aksomics

(Shanghai, China) was commissioned to perform tRF & tiRNA-seq library preparation, which includes 3' and 5' small RNA adapters, cDNA synthesis and library PCR amplification. The prepared tRF and tiRNA-seq libraries were subsequently quantified using the Agilent Bio-Analyzer 2100 and sequenced with an Illumina sequencer. The high-throughput sequencing results have been uploaded to the GEO database (GSE278659).

## 2.3 Extraction of total RNA, reverse transcription into cDNA, and quantitative RT-PCR

RNA was extracted from serum and cells using TRIzol reagent (Invitrogen, Shanghai, China). First-strand cDNA synthesis was carried out using the miRNA 1st Strand cDNA Synthesis Kit (by stem-loop) (Vazyme, Nanjing, China), followed by PCR amplification using miRNA Universal SYBR qPCR Master Mix (Vazyme, Nanjing, China).

The primer sequences used for tsRNA-49:73-Glu-CTC were as follows:

Forward (F): AAGAAGACGGGTTCGATTCCCG

Reverse (R): ATCCAGTGCAGGGTCCGAGG

Reverse Transcription (RT): GTCGTATCCAGTGCAGGGTCCGAGGTATTCGCACTGGATACGACGTTCCC

U6 small nuclear RNA was used as the internal reference gene, with the following primer sequences:

Forward (F): GCTTCGGCAGCACATATACTAAAAT

Reverse (R): CGCTTCACGAATTTGCGTGTCAT

Primer sequences are all synthesized by Sangon Biotech (Shanghai, China).

## 2.4 Functional analysis

**2.4.1 Cell transfection.** Based on the sequence of tsRNA-49-73-Glu-CTC, tsRNA-49-73-Glu-CTC Negative Control (NC) and inhibitor were synthesized by Sangon Biotech (Shanghai, China). The inhibitor NC sequence is 5'-CAGUACUUUUGUGUAGUACAA-3', and the tsRNA-49-73-Glu-CTC inhibitor sequence is 5'-GUUCCCUGACCGGGAAUCGAACCCG-3'. The solutions were prepared at a concentration of 20 μM, following the manufacturer's instructions. Transfection into A549 cells was carried out using Lipofectamine 2000 transfection reagent (ThermoFisher Scientific, MA, USA) at the recommended concentrations (15 pmol/well in 24-well plates). Functional assays were performed 24 to 48 hours post-transfection.

**2.4.2 Cell Counting Kit-8.** Transfected A549 cells were uniformly seeded into 96-well plates at a density of 1,000 cells per well. The CCK-8 reagent was subsequently added on day 1, 2, 3, 4, and 5 to measure the optical density (OD) values.

**2.4.3 Migration analysis.** Cells transfected with tsRNA-49-73-Glu-CTC NC or inhibitor were seeded into 6-well plates. Once the cells achieved 80% confluence, a scratch assay was conducted. The cells were subsequently cultured in media devoid of antibiotics, without serum, or with less than 1% serum. Scratch areas were measured at 0 h and 48 h, and the change in area was calculated to evaluate cell migration.

## 2.5 Statistical analysis

The diagnostic capability of candidate tsRNAs for NSCLC was assessed using receiver operating characteristic (ROC) curves [36], with the Youden index assisting in determining

the cutoff values. Data analysis was performed using SPSS (IBM SPSS Statistics 26.0; SPSS, Chicago, IL, USA) and GraphPad Prism 10 (GraphPad Software 10.1.2, San Diego, CA, USA). Data are presented as mean ± SE. Where appropriate, t-tests were used for data comparison. Pie charts, stacked bar charts, scatter plots, volcano plots, and heat maps were generated using the R package. A probability (P) value < 0.05 was considered statistically significant.

## 2.6 Approval and consent for ethics participation

The study protocol was approved by the Ethics Committee of the Affiliated Qingdao Central Hospital of the University of Rehabilitation (KY202415201) on 18/09/2024. The study was conducted in accordance with the ethical principles for medical research involving human subjects as described in the Declaration of Helsinki. Written informed consent was obtained from all participants prior to their inclusion in the study. All patient information has been replaced with identification codes, and no personal or identifiable data has been revealed.

## 3 Results

### 3.1 TsRNA expression profiles in NSCLC patients and healthy individuals

Serum samples were collected from three untreated NSCLC patients and three healthy individuals undergoing routine physical examinations. High-throughput sequencing was conducted on these samples, resulting in the identification of 3,522 types of tsRNAs (Fig 1A). The majority of the tsRNAs detected in the serum of both NSCLC patients and healthy individuals were novel discoveries, not previously recorded in common RNA databases (Fig 1C, D). The analysis revealed that 786 highly expressed tsRNAs (CPM > 20) were present in both NSCLC patients and healthy individuals, while 1,014 tsRNAs were specifically expressed in the serum of NSCLC patients, and only 195 tsRNAs were expressed in the serum of healthy individuals (Fig 1B). Further classification of tsRNAs showed that glutamate (Glu)-related tsRNAs were the most abundant. Moreover, we observed that while the types of tRNAs did not differ significantly between NSCLC patients and healthy individuals, there were notable differences in expression levels (Fig 1E, F).

### 3.2 Differential expression of tsRNAs

Further analysis revealed that 786 tsRNAs were differentially expressed in the serum of NSCLC patients compared to healthy individuals (Fig 2A). Using a fold change threshold of ≥ 1.5 and a p-value of ≤ 0.05, we identified 18 tsRNAs that were significantly upregulated and 15 tsRNAs that were significantly downregulated in cancer patients (Fig 2B, C). Subsequently, we selected the five most differentially expressed tsRNAs—tsRNA-49:73-Glu-CTC, tsRNA-19:32-Lys-CTT, tsRNA-18:32-Lys-CTT, tsRNA-14:32-Lys-CTT, and tsRNA-16:31-Lys-CTT— for RT-qPCR validation. RT-qPCR analysis of serum from the three NSCLC patients and three healthy individuals showed that tsRNA-49:73-Glu-CTC, tsRNA-19:32-Lys-CTT, and tsRNA-18:32-Lys-CTT were significantly upregulated in NSCLC patients compared to healthy individuals (P < 0.05) (Fig 2D, E, F). However, no significant differences were observed in the expression levels of tsRNA-14:32-Lys-CTT and tsRNA-16:31-Lys-CTT (S1 Fig).

### 3.3 Secondary structure prediction

The secondary structures of tsRNA-49:73-Glu-CTC, tsRNA-19:32-Lys-CTT, and tsRNA-18:32-Lys-CTT were predicted using the RNA secondary structure visualization tool (http://rna.tbi.univie.ac.at/forna/) (Fig 3). Given the requirements for subsequent experiments and the intrinsic stability of tsRNAs, we selected the longer and more structurally stable tsRNA-49:73-Glu-CTC as the target for further investigation.

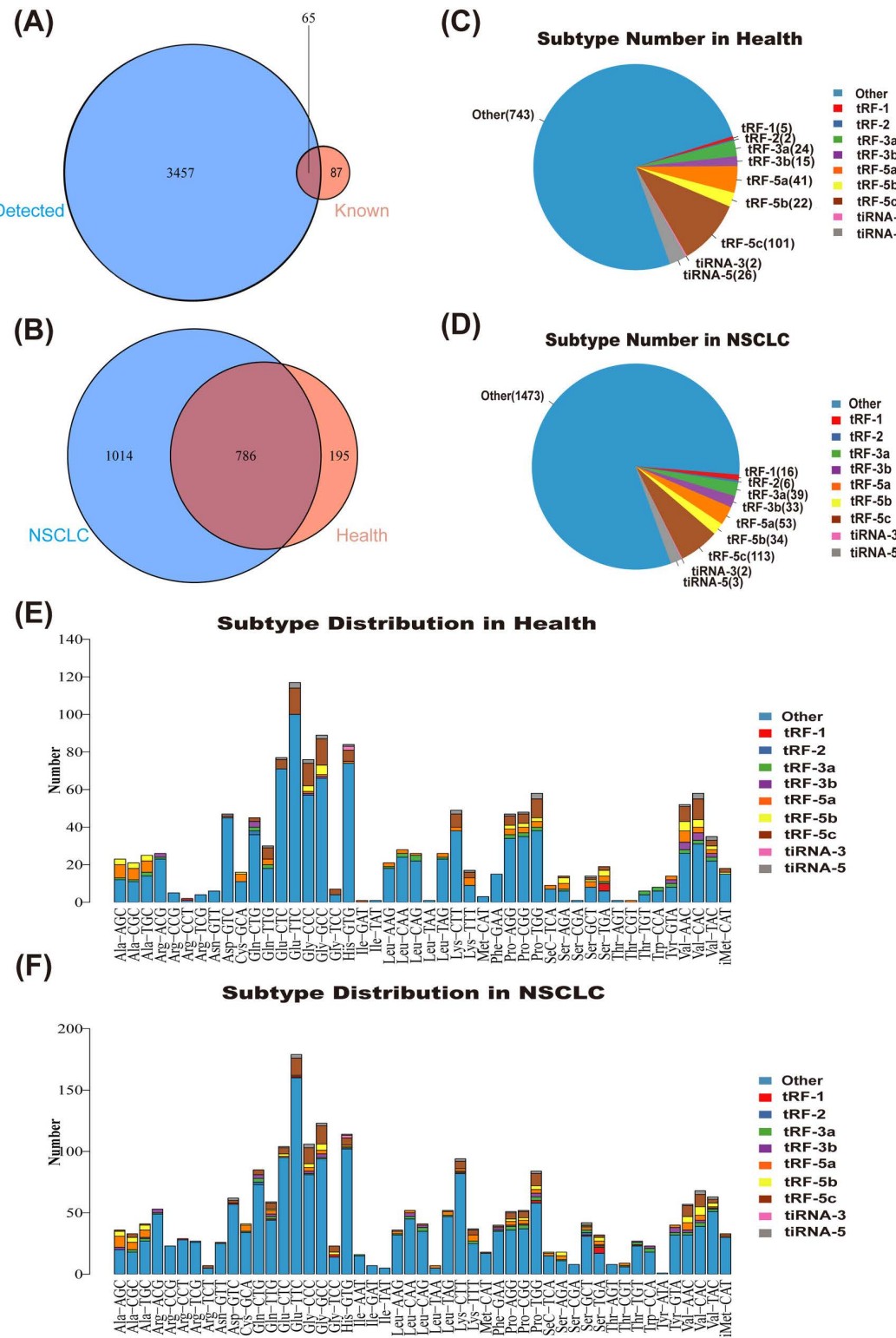

**Fig 1. Serum tsRNA expression profiles in NSCLC patients.** (A) Venn diagram showing the number of known and detected tsRNAs. (B) Number of tsRNAs expressed in the serum of NSCLC patients and healthy controls. (C, D) Distribution of tsRNA subtypes in NSCLC and healthy serum samples. (E, F) The number of subtype tsRNA against tRNA isodecoders. The X axes represents tRNA decoders and the Y axes shows the number of all subtype tsRNA against tRNA isodecoders. The color represents the subtype tsRNA.

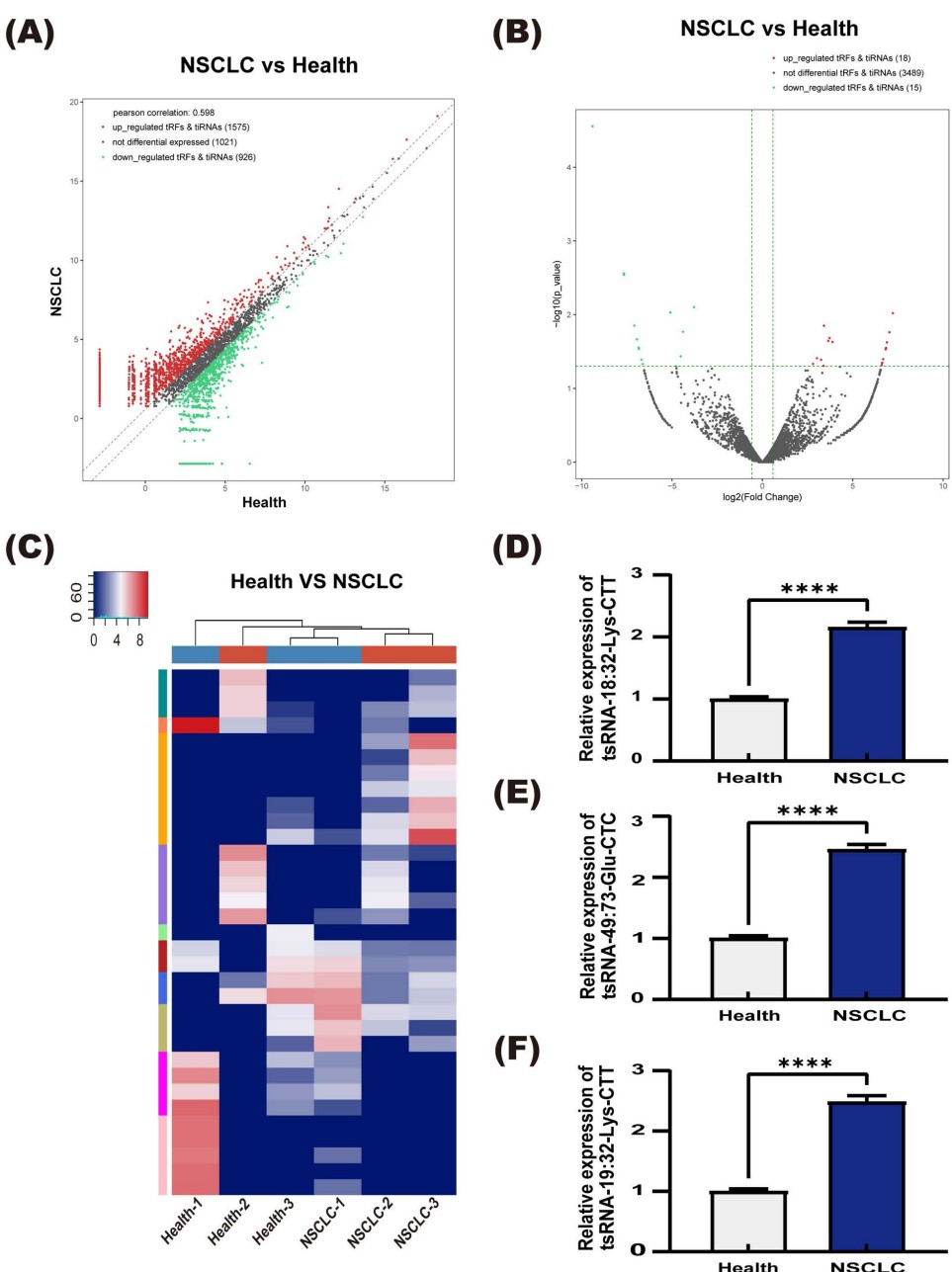

**Fig 2. Differentially expressed tsRNAs.** (A) Scatter plot of differentially expressed tsRNAs between NSCLC and healthy controls. The values on the X and Y axes represent the average CPM values for each group (log2 scaled). Red dots above the top line (upregulated) or green dots below the bottom line (downregulated) indicate tsRNAs with a fold change greater than 1.5 between the two comparison groups. Gray dots represent non-differentially expressed tsRNAs. (B) Volcano plot of significantly differentially expressed tsRNAs between NSCLC and healthy controls. (C) Heatmap of significantly differentially expressed tsRNAs between NSCLC and healthy controls. (D, E, F) Expression levels of tsRNA-49:73-Glu-CTC, tsRNA-19:32-Lys-CTT, and tsRNA-18:32-Lys-CTT in the serum of NSCLC patients and healthy controls, ****P < 0.0001.

### 3.4 Diagnostic value of tsRNA-49:73-Glu-CTC in serum

We conducted RT-qPCR analysis on the serum of 32 patients with NSCLC and 20 healthy controls. Compared to the control group, tsRNA-49:73-Glu-CTC was significantly upregulated in NSCLC patients, with a statistically significant difference (P < 0.05) (Fig 4A). Subsequently, we performed ROC analysis to assess the diagnostic value of tsRNA-49:73-Glu-CTC (Table 1). The results indicated that the area under the curve (AUC) for tsRNA-49:73-Glu-CTC was 0.785, with a sensitivity of 68.8% and a specificity of 84.2% (Fig 4B). In comparison to commonly used tumor markers (CEA, AUC = 0.589; NSE, AUC = 0.591; CYFRA, AUC = 0.560) (Fig 4C), tsRNA-49:73-Glu-CTC exhibited superior diagnostic value based on random samples from 72 NSCLC patients and 100 healthy individuals collected between April 2023 and July 2024. When combined with CEA, NSE, and CYFRA, the AUC for tsRNA-49:73-Glu-CTC was further enhanced, achieving a maximum of 0.870 when all four markers were utilized together (Fig 4D). These findings suggest that tsRNA-49:73-Glu-CTC has the potential to serve as a valuable diagnostic biomarker, and its combination with other clinical tumor markers can significantly enhance diagnostic accuracy.

### 3.5 Preliminary exploration of the function of tsRNA-49:73-Glu-CTC in NSCLC

Given the high expression of tsRNA-49:73-Glu-CTC in patients with NSCLC, we conducted a preliminary study to investigate its role in NSCLC development. Initially, we assessed the expression levels of tsRNA-49:73-Glu-CTC in normal lung epithelial cells (Beas-2b) and lung adenocarcinoma cells (A549). Our results revealed that tsRNA-49:73-Glu-CTC was significantly more expressed in A549 cells compared to Beas-2b cells, aligning with the changes observed in serum (Fig 5A). Following the transfection of A549 cells with a tsRNA-49:73-Glu-CTC inhibitor, we noted a substantial reduction in tsRNA-49:73-Glu-CTC expression, confirming the success of the inhibition (Fig 5B). Subsequent functional assays indicated that A549 cells transfected with the tsRNA-49:73-Glu-CTC inhibitor exhibited decreased proliferation compared to the control group, as evidenced by the CCK-8 assay (Fig 5C). Furthermore, the wound healing assay demonstrated that transfection with the tsRNA-49:73-Glu-CTC inhibitor hindered the migratory capacity of A549 cells (Fig 5D). These findings suggest that tsRNA-49:73-Glu-CTC may play a significant role in the proliferation and migration of NSCLC cells.

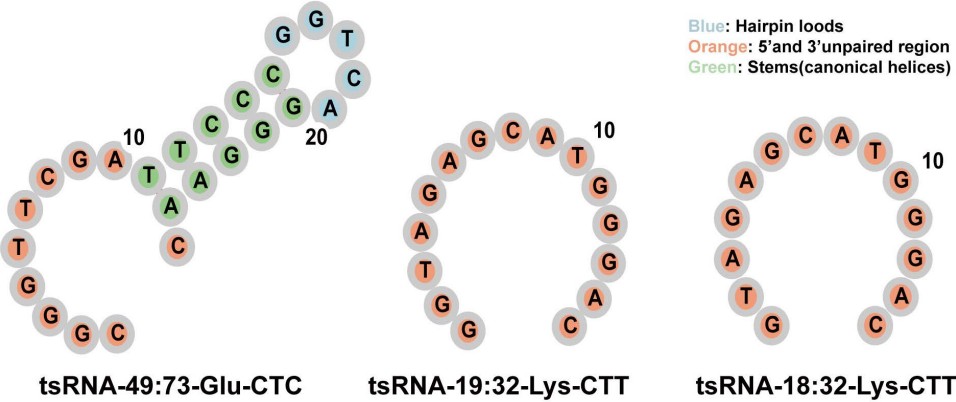

**Fig 3. Secondary structures of tsRNA-49:73-Glu-CTC, tsRNA-19:32-Lys-CTT, and tsRNA-18:32-Lys-CTT.**

## 4 Discussion

Lung cancer is the leading cause of both incidence and mortality rates worldwide, with 85% of cases classified as NSCLC. The five-year survival rate for this type of cancer is less than 20% [2]. One of the primary reasons for the high mortality rate associated with lung cancer is the absence of effective early diagnostic methods, which often result in late-stage diagnoses. This delay frequently leads to missed opportunities for optimal therapeutic intervention [8,9]. However, for early-stage lung cancer, the five-year survival rate can reach up to 70%

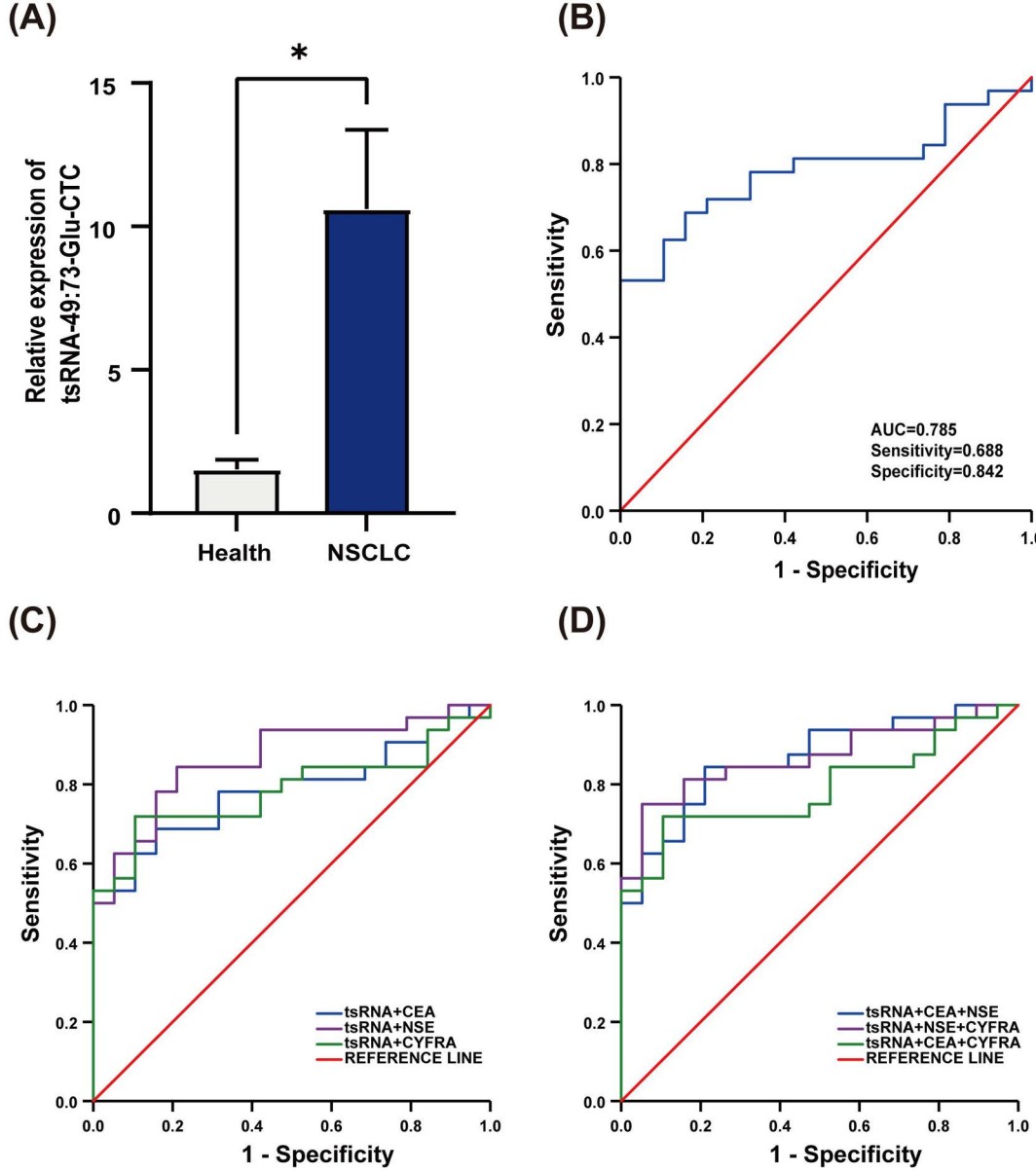

**Fig 4. Diagnostic value analysis of tsRNA-49:73-Glu-CTC in serum.** (A) Expression levels of tsRNA-49:73-Glu-CTC in the validation cohort. (B) ROC curve analysis of tsRNA-49:73-Glu-CTC. (C) ROC curve analysis of tsRNA-49:73-Glu-CTC combined with CEA, NSE, and CYFRA individually. (D) ROC curve analysis of tsRNA-49:73-Glu-CTC combined with CEA, NSE, and CYFRA. AUC: Area Under the Curve; NSE: Neuron-Specific Enolase; CYFRA: Cytokeratin 19 Fragment; tsRNA: tsRNA-49:73-Glu-CTC. * P < 0.05.

with treatment [37]. Therefore, early diagnosis and intervention are essential for the effective management of lung cancer. Among the various diagnostic approaches being explored, liquid biopsy has emerged as a promising tool due to its ability to detect molecular biomarkers in body fluids, offering new avenues for early cancer detection. Liquid biopsy presents several advantages, including its non-invasive nature, easy accessibility, and high clinical practicality [38]. Serum, as a commonly used diagnostic sample, is devoid of anticoagulants when compared to plasma, which leads to fewer interfering factors and enhanced stability [39]. Studies have identified tRF 17 18VBY9M in serum as a potential diagnostic biomarker for gastric cancer, demonstrating its ability to effectively differentiate between stage I/II and stage III/IV gastric cancer [40]. There are studies investigating the use of serum tsRNAs as molecular diagnostic biomarkers in various cancers, including pancreatic cancer, breast cancer, and renal cell carcinoma [41–43]. Therefore, this study aimed to address the critical question of whether potential molecular diagnostic biomarkers for NSCLC can be identified in serum.

Initially, this study employed high-throughput sequencing on serum samples from three pairs of NSCLC patients and healthy controls to identify tsRNA-49:73-Glu-CTC, which demonstrated significant differential expression in the serum of NSCLC patients. The sample size was subsequently expanded, and the elevated expression of tsRNA-49:73-Glu-CTC in the serum of NSCLC patients was validated using RT-qPCR. ROC curve analysis was conducted to compare tsRNA-49:73-Glu-CTC with commonly utilized clinical tumor markers (CEA, NSE, CYFRA), revealing that tsRNA-49:73-Glu-CTC achieved an AUC of 0.785, with a sensitivity of 68.8% and a specificity of 84.2%. This indicates a superior diagnostic performance compared to CEA, NSE, and CYFRA, highlighting its potential as a molecular biomarker. In this study, 88.9% of the NSCLC patients were in the early stages (IA-IIB), and it was observed that CEA, NSE, and Cyfra21-1 exhibited lower diagnostic capabilities for early-stage NSCLC compared to late-stage lung cancer, complicating the differentiation of early-stage lung cancer patients from healthy individuals [15]. Furthermore, tsRNA-49:73-Glu-CTC exhibits superior screening capability and, when combined with CEA, NSE, and CYFRA, significantly enhances the diagnostic performance of these biomarkers.

tsRNAs exhibit properties akin to microRNAs (miRNAs) and are implicated in the development and progression of cancer [21–23]. For instance, tRFdb-3013a/b can target and regulate the expression of ST3GAL1 in colon adenocarcinoma, thereby exerting

**Table 1. ROC curve analysis of individual and combined groups.**

| | AUC | SEN | SPE | cut off | 95% CI | |
|---|---|---|---|---|---|---|
| | | | | | Lower-bound | Upper-bound |
| tsRNA-49:73-GLU-CTC | 0.785 | 0.688 | 0.842 | 1.958 | 0.660 | 0.909 |
| CEA | 0.589 | 0.688 | 0.632 | 2.205 | 0.420 | 0.757 |
| NSE | 0.591 | 0.375 | 0.947 | 15.900 | 0.436 | 0.747 |
| CYFRA | 0.560 | 0.719 | 0.474 | 1.370 | 0.392 | 0.728 |
| tsRNA-49:73-GLU-CTC⁺CEA | 0.785 | 0.688 | 0.842 | 0.550 | 0.661 | 0.908 |
| tsRNA-49:73-GLU-CTC⁺NSE | 0.865 | 0.844 | 0.789 | 0.503 | 0.766 | 0.964 |
| tsRNA-49:73-GLU-CTC⁺CYFRA | 0.786 | 0.719 | 0.895 | 0.541 | 0.661 | 0.911 |
| tsRNA-49:73-GLU-CTC⁺CEA+NSE | 0.865 | 0.844 | 0.789 | 0.523 | 0.767 | 0.963 |
| tsRNA-49:73-GLU-CTC⁺CEA+CYFRA | 0.868 | 0.75 | 0.947 | 0.575 | 0.771 | 0.966 |
| tsRNA-49:73-GLU-CTC⁺NSE+CYFRA | 0.789 | 0.719 | 0.895 | 0.555 | 0.666 | 0.913 |
| tsRNA-49:73-GLU-CTC⁺CEA+NSE⁺CYFRA | 0.870 | 0.75 | 0.947 | 0.585 | 0.774 | 0.966 |

AUC: Area Under the Curve; SEN: Sensitivity; SPE: Specificity; CI: Confidence Interval.

antitumor effects [44]. tiRNA-Val-CAC-2 is significantly upregulated in metastatic pancreatic cancer lesions, promoting metastasis by enhancing FUBP1 stability and upregulating c-MYC transcription [45]. A specific group of tRFs functionally binds to the oncogenic RNA-binding protein YBX1, displacing YBX1 from its target transcripts and suppressing breast cancer progression [46]. Additionally, several studies have investigated the mechanisms of tsRNAs in lung cancer. For example, tsRNA-5001a can inhibit the antitumor function of the tumor suppressor gene GADD45G by reducing its stability, which in turn enhances tumor proliferation [47]. AS-tDR-007333 can enhance the proliferation of NSCLC cells by activating the interaction between HSPB1 and MED29, as well as by

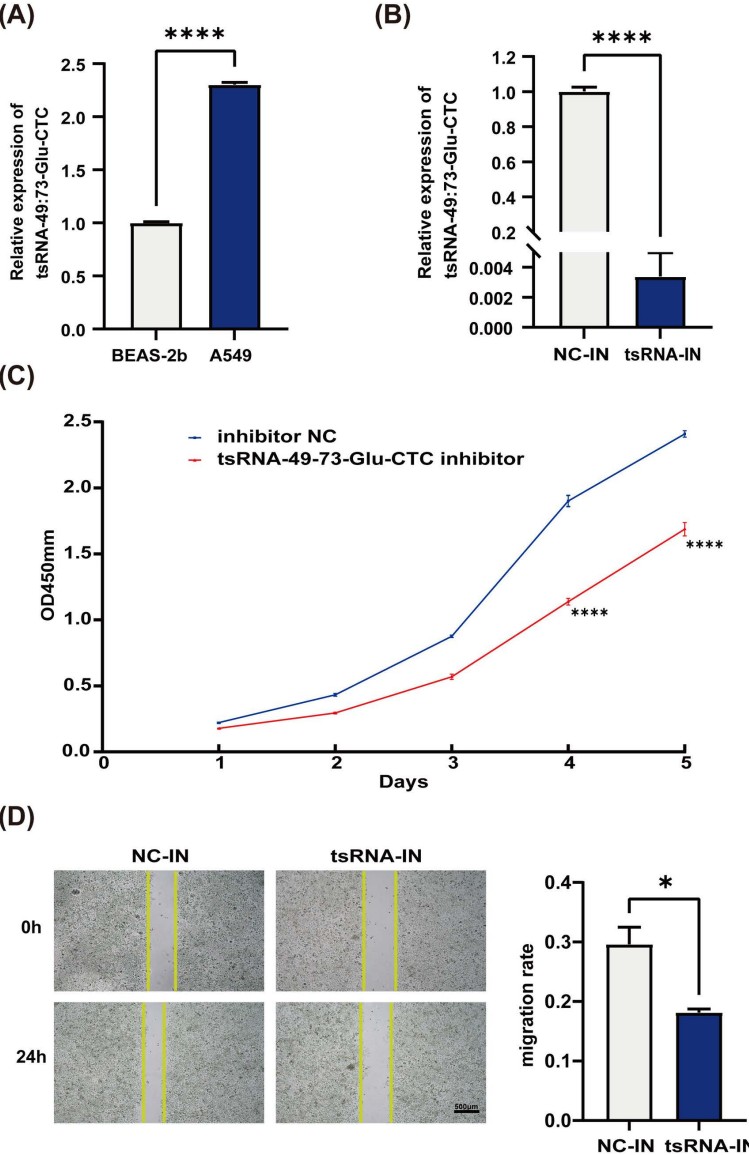

**Fig 5. Preliminary functional exploration of tsRNA-49:73-Glu-CTC.** (A) Expression levels of tsRNA-49:73-Glu-CTC in BEAS-2B and A549 cells. (B) Expression of tsRNA-49:73-Glu-CTC in A549 cells after transfection with tsRNA-49:73-Glu-CTC inhibitor. (C) CCK-8 cell proliferation assay. (D) Cell scratch assay. NC-IN: inhibitor NC; tsRNA-IN: tsRNA-49:73-Glu-CTC inhibitor. * $P < 0.05$, **** $P < 0.0001$.

interacting with ELK4 to modulate the transcription of the MED29 promoter. This finding reveals a novel mechanism through which transfer RNA fragments (tRFs) interact with HSPB1 and ELK4 to regulate NSCLC cell proliferation [48]. This study found that tsRNA-49:73-Glu-CTC was expressed at significantly higher levels in A549 cells compared to normal lung epithelial cells. To investigate the effect of tsRNA-49:73-Glu-CTC on NSCLC, we transfected A549 cells with a tsRNA-49:73-Glu-CTC inhibitor. The results indicated that transfected cells exhibited reduced proliferation and diminished migratory ability, suggesting that tsRNA-49:73-Glu-CTC may play a role in the proliferation and migration processes associated with NSCLC. However, due to the small sample size and limited geographic scope of this study, future research should incorporate larger sample sizes and broader geographic representation to validate these findings. Additionally, the function of tsRNA-49:73-Glu-CTC has only been preliminarily examined, necessitating further research to elucidate its underlying mechanisms.

## 5  Conclusion

This study demonstrates that tsRNA-49:73-Glu-CTC is highly expressed in the serum of patients with NSCLC, indicating its potential as a novel non-invasive biomarker. Furthermore, it is implicated in the proliferation and migration of A549 cells, thereby opening new avenues for research into the pathogenesis of NSCLC.

## Supporting information

**S1 Table.  Patient characteristics.**
(DOCX)

**S1 Fig.  Expression levels of tsRNA-14:32-Lys-CTT and tsRNA-16:31-Lys-CTT in the serum of NSCLC patients and healthy controls.**
(TIF)

## Acknowledgments

We sincerely acknowledge the support from Qingdao Central Hospital and the invaluable guidance of our mentor. Special thanks to the team members for their collaborative spirit. We also appreciate the reviewers for their patience and valuable feedback.

## Author contributions

**Conceptualization:** Xiaofeng Mu.

**Writing – original draft:** Chenyu Li, Shenjie Zhong.

**Writing – review & editing:** Juan Chen, Xiaofeng Mu.

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
