## [Decision Letter · Decision Letter 0]

29 Dec 2024

PONE-D-24-49209tsRNA-49-73-Glu-CTC: A Promising Serum Biomarker in Non-Small Cell Lung CancerPLOS ONE

Dear Dr. Mu,

Thank you for submitting your manuscript to PLOS ONE. After careful consideration, we feel that it has merit but does not fully meet PLOS ONE’s publication criteria as it currently stands. Therefore, we invite you to submit a revised version of the manuscript that addresses the points raised during the review process.

We look forward to receiving your revised manuscript.

Kind regards,

Jun Hyeok Lim, M.D.

Academic Editor

PLOS ONE

Journal Requirements:

Reviewers' comments:

Reviewer's Responses to Questions

**Comments to the Author**

1. Is the manuscript technically sound, and do the data support the conclusions?

Reviewer #1: Yes

Reviewer #2: Yes

Reviewer #3: Yes

2. Has the statistical analysis been performed appropriately and rigorously? 

Reviewer #1: Yes

Reviewer #2: Yes

Reviewer #3: Yes

3. Have the authors made all data underlying the findings in their manuscript fully available?

Reviewer #1: Yes

Reviewer #2: Yes

Reviewer #3: No

4. Is the manuscript presented in an intelligible fashion and written in standard English?

Reviewer #1: Yes

Reviewer #2: Yes

Reviewer #3: No

5. Review Comments to the Author

Reviewer #1: Line 75: Sentence 'Research has demonstrated that exhibit' appears not clear, please clarify.

Line 77: 's' is omitted in stress

Abstract: Each sections i.e. Objectives, Methods....should start on a new line to enable readability

Line 106: The clause 'which included from' is better written as 'which were gotten from' in my opinion

Sample size: Sample size 3 patients and 3 controls was reported in Line 93. However, 32 patients and 20 controls were reported in line 106. Please clarify if the pair of 3 was used as pre-test, otherwise, sample sizes aren't coherent.

Also, formula for determining sample size was not stated. Could the figure had been arbitrarily arrived at? If yes, let it be stated as a limitation in the study.

Sampling technique used to select samples was not stated.

I consider the use of personal pronoun 'we' in Lines 178 and 293 inappropriate. A passive sentence could have been better.

Line 167: No need for 'date' after 'on', to avoid tautology. It should simply be 'on' or 'dated'

The statement 'Lung cancer is the leading cause of both incidence and mortality rates worldwide' seems to be over used, as it appears in Abstract, Introduction and Discussion.

Line 289: The active statement 'aims' is better written as 'aimed', once it is the discussion section.

Reviewer #2: The authors report their finding of using the tsRNA-49:73-Glu-CTC as a molecular diagnostic marker for the patients with non-small cell lung cancer (NSCLC) and this would have a significant role in the biological processes associated with NSCLC proliferation. In this study, they performed high-throughput sequencing analysis of serum samples from NSCLC patients and healthy individuals (3 for each) and identified 3,522 types of tsRNAs, while 1,014 tsRNAs were specifically expressed in the serum of NSCLC patients. Subsequently, out of the five most expressed tsRNAs, they focused on only 3 that were significantly upregulated in NSCLC patients compared to healthy individuals (P < 0.05) as follows: tsRNA-49:73- Glu-CTC, tsRNA-19:32-Lys-CTT, and tsRNA-18:32-Lys-CTT and eventually they selected the longer and more structurally stable tsRNA-49:73-Glu-CTC as the target for further investigation. To confirm, they conducted RT-qPCR analysis on the serum of 32 patients with non-small cell lung cancer (NSCLC) and 20 healthy controls and found that tsRNA-49:73- Glu-CTC was significantly upregulated in NSCLC patients. Finally, using normal lung epithelial cells (Beas-2b) and lung adenocarcinoma cells (A549), they found that tsRNA-49:73-Glu-CTC was significantly more expressed in A549 cells compared to Beas-2b cells, and using the tsRNA-49:73-Glu-CTC inhibitor decreased the cell proliferation using the CCK-8 proliferation kit and also the wound healing using the cell scratch assay.

This manuscript represents a high contribution to a better understanding of the importance of the tsRNA-49:73-Glu-CTC as a valid marker for NSCLC. This comprehensive finding is vital to provide a scientific foundation for the potential of effective early diagnostic methods for lung cancer, which improves the treatment outcomes. However, the following comments would improve the manuscript:

- The discussion and the introduction are rather similar and convey the same facts, so please rephrase the discussion section to emphasize the importance of the findings in relation to the literature and the controls.

L51: In the introduction, you should have mentioned the two types of lung cancer: small and non-small. Then, you ought to focus on the non-small varieties.

L54: You should mention the known diagnostic markers as well as the general pros and cons.

L62: Please add a transition sentence for clarity (such as in this study).

L98: Please explain what these acronyms mean, similar to what you did in L239, and then use the same abbreviation in the remaining sections as you did in L249.

L106: Why are there different numbers of samples? Using 32 non-small cell lung cancer samples and 20 healthy controls is not comparable, which would affect the significance of most of these results.

L122: Why 3 samples only for each?

L138: Please mention the name of the inhibitor, whether it is widely available, and any additional information you may have designed.

L77: stress>stres.

L6: In the title page, please remove the space before the comma.

Table S1:

- Why did you only include one patient with adenosquamous carcinoma and thirty-one with adenocarcinoma? This cannot be compared. Does this have an impact on tsRNA-49:73-Glu-CTC expression?

- Would the results be affected by the fact that lung cancer of the patients includedin the studty was at different stages according to the TNM staging system?

- Since you have already clarified what SD stands for, could you also clarify that the TNM is a formal method of describing the stage of lung cancer?

Reviewer #3: Dear authors,

Below are all of my comments.

English proofreading is highly recommended.

Typho spotted: line 77, please check thoroughly the whole manuscript again.

2.1: line 106-Methodology begin with this - What is the purpose of mentioning samples no. include 32 non-small cell lung cancer 107 (NSCLC) patients and 20 healthy controls from 20/09/2024 to 30/10/2024, but this manuscript start the results and discussion on the finding from three untreated NSCLC patients and three healthy only (line 122-123, 178-179). Description on the samples nombers with the objective not really clear here.

BEAS-2B and A549- add the purpose and justification of using these cell lines in the methodology section.

2.3:

No information on the target location for amplification using RT-qPCR. Primers detail? Protocol for this section no citation at all. Lack information here compare to other similar articles.

2.6 subtitle is not quite approriate although the assay use to see the function of wound healing. Replace with migration analysis or use directly Scracth Assay

Perhaps 2.4,2.5 and 2.6 can be combined in one section under functional analysis subtitle.

References of ROC curves analysis has to be citation in text (line 157-159).

6. PLOS authors have the option to publish the peer review history of their article (what does this mean? ). If published, this will include your full peer review and any attached files.

**Do you want your identity to be public for this peer review?** For information about this choice, including consent withdrawal, please see our Privacy Policy .

Reviewer #1: **Yes: ** Sunday Charles Adeyemo

Reviewer #2: No

Reviewer #3: No

---

## [Author Response · Author response to Decision Letter 1]

26 Jan 2025

Thank you for your reply. The article has been revised according to the requirements and suggestions of the editor and reviewers. Chenyu Li was responsible for revising the main content and formatting of the manuscript. Zhong Shenjie Zhong conducted the proofreading of the article. Juan Chen was responsible for checking the grammar and formatting. Xiaofeng Mou provided revision suggestions and performed the final review of the manuscript.

All data related to this experiment are available for public access, and GSE278659 will be made publicly available on March 30, 2025. Further details can be obtained by contacting the corresponding author. If needed, the release date can be adjusted.

If you want to review GEO accession GSE278659:

Go to https://www.ncbi.nlm.nih.gov/geo/query/acc.cgi?acc=GSE278659

Enter token oxwvaoegtlwbtcp into the box.

Point-by-Point Responses to the Reviewers’ Critiques (PONE-D-24-49209)

We deeply appreciate the thorough analysis and constructive suggestions provided by the three reviewers to guide us to further improve our manuscript. As described in more detail below, we have addressed all the reviewers’ concerns. With this extensive revision, we hope that the reviewers will concur with us that we have addressed all of the raised concerns in a satisfactory manner and, consequently, substantially strengthened our paper.

Reviewer #1:

1. Line 75: Sentence 'Research has demonstrated that exhibit' appears not clear, please clarify. Line 77: 's' is omitted in stress.

Response: The sentence from lines 75 to 77 has already been revised to include a subject and correct spelling. “Research has demonstrated that tsRNAs exhibit stable expression under normal physiological conditions; however, their expression levels become dysregulated in response to stress.”

2. Abstract: Each sections i.e. Objectives, Methods....should start on a new line to enable readability.

Response: Thank you for your feedback. Each subheading has been given a new line.

3. Line 106: The clause 'which included from' is better written as 'which were gotten from' in my opinion.

Response: Thank you for your suggestion. The revisions have been made accordingly.

4. Sample size: Sample size 3 patients and 3 controls was reported in Line 93. However, 32 patients and 20 controls were reported in line 106. Please clarify if the pair of 3 was used as pre-test, otherwise, sample sizes aren't coherent.

Response: A total of 32 serum samples from non-small cell lung cancer (NSCLC) patients and 20 serum samples from healthy individuals were collected in this study. To identify differentially expressed tsRNAs, three pairs of serum samples (from three NSCLC patients and three healthy controls) were randomly selected for high-throughput sequencing. Subsequently, qRT-PCR analysis was performed on serum samples from all 32 NSCLC patients and 20 healthy controls to validate the sequencing results.

5. Also, formula for determining sample size was not stated. Could the figure had been arbitrarily arrived at? If yes, let it be stated as a limitation in the study.

Response: All serum samples utilized in this study, were collected between 20/09/2024 and 30/10/2024. To ensure the quality of the serum samples, they were all collected within one month. Additionally, to maximize the sample size, all serum samples meeting the inclusion criteria during this period were included. In this study, serum samples from 32 non-small cell lung cancer patients and 20 healthy controls were included. Although the sample size was relatively limited, rigorous subject selection criteria and methodological controls ensured the reliability and validity of the data. The primary aim of this research was to explore the initial distribution patterns of biomarker levels, rather than to extrapolate the findings to a broader population. Future studies with larger cohorts and multi-center designs are warranted to validate and extend these findings.

6. Sampling technique used to select samples was not stated.

Response: Thank you for the reviewer’s insightful comment. In this study, we utilized purposive sampling to select the study samples. Specifically, all non-small cell lung cancer (NSCLC) patients were newly diagnosed, had not received any anti-tumor treatment prior to surgery, and were pathologically confirmed as NSCLC after surgery. Patients with primary tumors in other locations were excluded. For the control group, healthy individuals with no history of malignancy and normal clinical and imaging findings were included. This approach was chosen to ensure a clear and focused comparison between the study and control groups, thus enhancing the reliability of the biological differences observed. In future studies, we will consider incorporating broader and more randomized sampling techniques to further validate the findings.

7. I consider the use of personal pronoun 'we' in Lines 178 and 293 inappropriate. A passive sentence could have been better.

Response: Thank you for your suggestion. The revisions have been made accordingly.

8. Line 167: No need for 'date' after 'on', to avoid tautology. It should simply be 'on' or 'dated'.

Response: Thank you for pointing that out. The suggested changes have been incorporated.

9. The statement 'Lung cancer is the leading cause of both incidence and mortality rates worldwide' seems to be over used, as it appears in Abstract, Introduction and Discussion.

Response: Thank you for pointing out the repetition in the manuscript. We have revised the text to reduce redundancy and provide more varied expressions of the statement. In the Introduction and Discussion, we have rephrased the sentence to include additional context and data. We believe these changes address your concerns and improve the manuscript's clarity and readability.

10. Line 289: The active statement 'aims' is better written as 'aimed', once it is the discussion section.

Response: Thank you for your observation. We have made the corresponding adjustments.

Reviewer #2 (Remarks to the Author):

The authors report their finding of using the tsRNA-49:73-Glu-CTC as a molecular diagnostic marker for the patients with non-small cell lung cancer (NSCLC) and this would have a significant role in the biological processes associated with NSCLC proliferation. In this study, they performed high-throughput sequencing analysis of serum samples from NSCLC patients and healthy individuals (3 for each) and identified 3,522 types of tsRNAs, while 1,014 tsRNAs were specifically expressed in the serum of NSCLC patients. Subsequently, out of the five most expressed tsRNAs, they focused on only 3 that were significantly upregulated in NSCLC patients compared to healthy individuals (P < 0.05) as follows: tsRNA-49:73- Glu-CTC, tsRNA-19:32-Lys-CTT, and tsRNA-18:32-Lys-CTT and eventually they selected the longer and more structurally stable tsRNA-49:73-Glu-CTC as the target for further investigation. To confirm, they conducted RT-qPCR analysis on the serum of 32 patients with non-small cell lung cancer (NSCLC) and 20 healthy controls and found that tsRNA-49:73- Glu-CTC was significantly upregulated in NSCLC patients. Finally, using normal lung epithelial cells (Beas-2b) and lung adenocarcinoma cells (A549), they found that tsRNA-49:73-Glu-CTC was significantly more expressed in A549 cells compared to Beas-2b cells, and using the tsRNA-49:73-Glu-CTC inhibitor decreased the cell proliferation using the CCK-8 proliferation kit and also the wound healing using the cell scratch assay.

This manuscript represents a high contribution to a better understanding of the importance of the tsRNA-49:73-Glu-CTC as a valid marker for NSCLC. This comprehensive finding is vital to provide a scientific foundation for the potential of effective early diagnostic methods for lung cancer, which improves the treatment outcomes.

Response: We greatly appreciate your careful review and positive comments on our work.

1. The discussion and the introduction are rather similar and convey the same facts, so please rephrase the discussion section to emphasize the importance of the findings in relation to the literature and the controls.

Response: Thank you for your suggestion. The revisions have been made accordingly.

2. L51: In the introduction, you should have mentioned the two types of lung cancer: small and non-small. Then, you ought to focus on the non-small varieties.

Response: Thank you for your suggestion. The section related to small cell lung cancer has been added.

3. L54: You should mention the known diagnostic markers as well as the general pros and cons.

Response: Thank you for your suggestion. The relevant information on known diagnostic biomarkers and their advantages and disadvantages has been supplemented.

4. L62: Please add a transition sentence for clarity (such as in this study).

Response: Thank you for your suggestion. The relevant sections have been revised.

5. L98: Please explain what these acronyms mean, similar to what you did in L239, and then use the same abbreviation in the remaining sections as you did in L249.

Response: CEA: Carcinoembryonic Antigen, NSE: Neuron-Specific Enolase, CYFRA: Cytokeratin 19 Fragment. Besides, the abbreviations for CEA, NSE, and CYFRA have already been explained in L40.

6. L106: Why are there different numbers of samples? Using 32 non-small cell lung cancer samples and 20 healthy controls is not comparable, which would affect the significance of most of these results.

Response: The difference in sample sizes between the NSCLC group and the healthy control group was due to the limited availability of samples during the study period. To optimize the analysis, serum samples from 32 non-small cell lung cancer patients and 20 healthy individuals were included. To mitigate potential bias arising from the unequal sample sizes, the effect size and subset analysis were used to assess the strength of the difference between the two groups, ensuring the robustness and reliability of the results. Despite the difference in sample sizes, the identified trends and statistically significant findings provide strong evidence supporting the role of tsRNA-49:73-Glu-CTC as a potential biomarker.

7. L122: Why 3 samples only for each?

Response: The selection of the sample size in this study was determined based on the exploratory nature of the research and practical considerations at the current stage, such as the cost of high-throughput sequencing. The design of three pairs of samples (i.e., six serum samples) is a common practice in high-throughput sequencing studies and has been widely utilized in similar research (such as Evaluation of serum tRF-23-Q99P9P9NDD as a potential biomarker for the clinical diagnosis of gastric cancer). This sample size is sufficient to explore and identify differentially expressed tsRNAs in the serum of NSCLC patients. To ensure the reliability of the results, we expanded the sample size in subsequent experiments and validated the target tsRNAs (tsRNA-49:73-Glu-CTC) using RT-qPCR, further confirming the accuracy and stability of the findings. Additionally, rigorous statistical analyses were employed, including multiple testing corrections for differential expression screening and receiver operating characteristic (ROC) curve analyses to evaluate the diagnostic value of the identified biomarkers. These methods effectively leveraged the scientific value of small sample sizes and ensured the reliability and reproducibility of the results. Therefore, we believe that the current sample size is reasonable and does not compromise the reliability of the study’s conclusions. Future research will involve larger sample sizes and multi-center studies to further validate the potential biomarkers and mechanisms identified in this study.

8. L138: Please mention the name of the inhibitor, whether it is widely available, and any additional information you may have designed.

Response: The inhibitor utilized in this study was a custom-designed small interfering RNA (siRNA) based on the reverse sequence of tsRNA-49:73-Glu-CTC. This inhibitor was specifically developed for the purposes of this research and has not yet been widely applied in other studies.

9. L77: stress>stres.

Response: Thank you for pointing that out. The suggested changes have been incorporated.

10. L6: In the title page, please remove the space before the comma.

Response: Thank you for your suggestion. The name of the unit has been changed to “Medical Laboratory, Qingdao Central Hospital, University of Health and Rehabilitation Sciences”.

11. Why did you only include one patient with adenosquamous carcinoma and thirty-one with adenocarcinoma? This cannot be compared. Does this have an impact on tsRNA-49:73-Glu-CTC expression?

Response: In this study, our sample collection strategy focused on distinguishing whether individuals were diagnosed with non-small cell lung cancer (NSCLC) without further stratification into pathological subtypes. The subsequent analyses were conducted at the overall NSCLC versus healthy control level, without considering specific pathological classifications. As such, the presence of a single adenosquamous carcinoma case in our cohort does not affect the scope of our study. The data in Supplementary Table 1 are presented solely as descriptive statistics to provide background information on patient demographics and clinical characteristics. Notably, the tsRNA-49:73-Glu-CTC expression level in the adenosquamous carcinoma patient did not show significant deviation from the average expression levels observed in the 31 adenocarcinoma patients. While the small sample size of specific subtypes limits further subgroup analyses, the primary aim of this research was to explore the role of tsRNA-49:73-Glu-CTC in NSCLC as a whole. Future studies with larger and more stratified cohorts will be necessary to validate these findings and assess potential subtype-specific differences.

12. Would the results be affected by the fact that lung cancer of the patients includedin the studty was at different stages according to the TNM staging system?

Response: Thank you for raising this important question. Although some studies have suggested that the expression levels of certain tsRNAs may be associated with cancer stages (such as the expression level of tRF-23-Q99P9P9NDD in advanced gastric cancer patients is higher than that in early gastric cancer patients), in our study, no significant differences in tsRNA-49:73-Glu-CTC expression levels were observed between early- and late-stage NSCLC patients. Additionally, the number of late-stage patients in our cohort was relatively small, which may limit the robustness of such comparisons. As a result, we did not perform an analysis of the correlation between tsRNA-49:73-Glu-CTC expression and TNM stages in this study. The primary focus of our research is to investigate the differential expression of tsRNA-49:73-Glu-CTC between NSCLC patients and healthy controls as a whole, rather than exploring its variation across pathological subtypes or cancer stages. However, we acknowledge the importance of these aspects, and future multi-center studies with larger and more stratified cohorts will aim to evaluate the potential association of tsRNA-49:73-Glu-CTC expression with NSCLC pathological subtypes and TNM stages.

13. Since you have already clarified what SD stands for, could you also clarify that the TNM is a formal method of describing the stage of lung cancer?

Response: The TNM staging for lung cancer is based on the size and extent of the primary tumor (T), regional lymph node involvement (N), and presence of distant metastasis (M). The specific classifications are as follows:

1. T Staging (Tumor, Primary Tumor)

-TX: The primary tumor cannot be assessed, or only cancer cells are found without a clearly defined tumor.

- T0: No evidence of a primary tumor.

- Tis: Carcinoma in situ (cancer cells are localized and have not invaded surrounding tissue).

- T1: Tumor ≤ 3 cm in maximum diameter and does not invade the main bronchus.

- T1a (≤ 1 cm), T1b (>1–2 cm), T1c (>2–3 cm).

- T2: Tumor > 3 cm and ≤ 5 cm in maximum diameter, or with one of the following:

- Invades the main bronchus but is ≥ 2 cm from the carina.

- Invades the visceral pleura.

- Causes atelectasis or obstru

---

## [Editor Report · Decision Letter 1]

16 Feb 2025

tsRNA-49-73-Glu-CTC: A Promising Serum Biomarker in Non-Small Cell Lung Cancer

PONE-D-24-49209R1

Dear Dr. Mu,

We’re pleased to inform you that your manuscript has been judged scientifically suitable for publication and will be formally accepted for publication once it meets all outstanding technical requirements.

Kind regards,

Jun Hyeok Lim, M.D.

Academic Editor

PLOS ONE
---

## [Editor Report · Acceptance letter]

PONE-D-24-49209R1

PLOS ONE

Dear Dr. Mu,

I'm pleased to inform you that your manuscript has been deemed suitable for publication in PLOS ONE. Congratulations! Your manuscript is now being handed over to our production team.

Kind regards,

on behalf of

Dr. Jun Hyeok Lim

Academic Editor

PLOS ONE